# In vitro functional analysis of gRNA sites regulating assembly of hepatitis B virus

Nikesh Patel [1✉], Sam Clark [2], Eva U. Weiß[2,4], Carlos P. Mata[1,5], Jen Bohon [3,6], Erik R. Farquhar [3], Daniel P. Maskell[1], Neil A. Ranson [1], Reidun Twarock[2] & Peter G. Stockley [1✉]

The roles of RNA sequence/structure motifs, Packaging Signals (PSs), for regulating assembly of an HBV genome transcript have been investigated in an efficient in vitro assay containing only core protein (Cp) and RNA. Variants of three conserved PSs, within the genome of a strain not used previously, preventing correct presentation of a Cp-recognition loop motif are differentially deleterious for assembly of nucleocapsid-like particles (NCPs). Cryo-electron microscopy reconstruction of the $T = 4$ NCPs formed with the wild-type gRNA transcript, reveal that the interior of the Cp shell is in contact with lower resolution density, potentially encompassing the arginine-rich protein domains and gRNA. Symmetry relaxation followed by asymmetric reconstruction reveal that such contacts are made at every symmetry axis. We infer from their regulation of assembly that some of these contacts would involve gRNA PSs, and confirmed this by X-ray RNA footprinting. Mutation of the ε stem-loop in the gRNA, where polymerase binds in vivo, produces a poor RNA assembly substrate with Cp alone, largely due to alterations in its conformation. The results show that RNA PSs regulate assembly of HBV genomic transcripts in vitro, and therefore may play similar roles in vivo, in concert with other molecular factors.

[1] Astbury Centre for Structural Molecular Biology, University of Leeds, Leeds LS2 9JT, UK. [2] Departments of Biology and Mathematics & York Centre for Complex Systems Analysis, University of York, York YO10 5DD, UK. [3] CWRU Center for Synchrotron Biosciences, NSLS-II, Brookhaven National Laboratory, Upton, NY 11973, USA. [4] Present address: Institute of Molecular Infection Biology (IMIB), University of Würzburg, Josef-Schneider-Str. 2/D15, D-97080 Würzburg, Germany. [5] Present address: Electron and Confocal Microscopy Unit (UCCTs), National Centre for Microbiology (ISCIII). Majadahonda, Madrid, Spain. [6] Present address: Los Alamos National Laboratory, Los Alamos, NM 87545, USA. ✉email: fbsnpat@leeds.ac.uk; P.G.Stockley@leeds.ac.uk

epatitis B Virus (HBV) has infected over 2 billion people worldwide[1], ~240 million of whom are chronically infected after failing to clear an acute primary infection. Within this cohort, failure to suppress the virus leads eventually to liver failure, cirrhosis and cancer, resulting in ~700,000 deaths annually[2]. Despite an effective vaccine, over a million new infections occur every year[3,4]. For chronically infected patients therapy options are limited. Clinical therapy commonly uses nucleos(t)ide analogue inhibitors of viral polymerase (Pol) but this rarely leads to a cure and elicits rapid resistance mutations[5,6]. HBV represents one of the largest health challenges of any viral pathogen. A WHO Global Challenge has been established with the goal of making chronic infection treatable by 2030[7]. Novel curative therapies require improved mechanistic understanding of the HBV lifecycle.

The basis of the chronic infection is a covalently-closed, circular DNA (cccDNA) copy of the viral genome which persists in the nucleus as a chromatinized episome[8,9]. HBV is a para-retrovirus, i.e. a DNA virus that initially packages a positive-sense, single-stranded (ss) pre-genome, pgRNA[10,11], into a nucleocapsid (NC) composed of multiple Cp dimers (Fig. 1). These form $T = 3$ or $T = 4$ surface lattices, the latter being dominant. The 3200 bp long genome encodes four overlapping reading frames for polymerase (Pol); surface proteins (HBsAg); the cell regulatory factor protein X; and the core and pre-core proteins (HBcAg and HBeAg, respectively). Pol and Cp proteins are translated from the pgRNA, which also serves as the template for reverse transcription within the NC shell. The pgRNA is a 5′-capped, terminally redundant, poly-A tailed, mRNA transcript ~3500 nts long.

Previously, RNA SELEX[12,13] against recombinant full-length HBV Cp (183 aa long) dimers from strain NC_003977.1 was used to isolate aptamers whose sequences align with genomic sequences in the cognate pgRNA. Analysis of these genomic matching sites identifies their common features. Each site has the potential to form a stem-loop with a defined loop sequence motif, -RGAG-. Such sites are highly conserved across the pgRNAs of many strain variants. These HBV sites, as isolated RNA oligonucleotides, trigger in vitro sequence-specific NCP formation at nanomolar concentrations[12]. We propose that they act as Packaging Signals (PSs) helping to ensure faithful encapsidation of the pgRNA, by analogy with the mechanism regulating assembly of many bona fide ssRNA viruses[14–20].

Here we show that homologous PSs occur in similar genome locations in a commercially-available strain variant (JQ707375.1), which was not included in the previous analysis. These sites regulate in vitro assembly of $T = 3$ and $T = 4$ nucleocapsid-like particles (NCPs) in the context of a long genomic gRNA fragment lacking a 5′ cap and a poly-A tail. Regulation occurs at least in part through Cp recognition of the conserved loop -RGAG- motif of the PSs. Some of these contacts remain in the assembled particle. Symmetry expansion[21–24] of an icosahedrally-averaged ~3.2 Å resolution cryo-electron microscopy (cryo-EM) reconstruction of the reassembled $T = 4$ NCPs reveals multiple contacts at all symmetry axes between encapsidated density (the C-terminal arginine-rich domains (ARDs) of Cp's & gRNA) and the globular Cp shell. X-ray RNA footprinting (XRF)[25–27] confirms that the most conserved PS makes one such contact via its -RGAG- motif. Assembly in vitro occurs in the absence of viral polymerase which binds a stem-loop (ε) on the pgRNA[28], and post-translational modifications of the Cp. Both features assist NCP formation in vivo[28–32]. Mutation to prevent ε interacting with the distal φ site creates a genomic fragment that is considerably larger than both wild-type and PS mutant RNAs, as well as being a poor in vitro assembly substrate. These data highlight that PS-mediated assembly via the formation of multiple PS-Cp contacts promotes NCP-like formation, in the absence of other mechanisms regulating assembly.

## Results

**Identification of PS sites in HBV strain JQ707375.1.** Evolutionarily conserved PS sites in the pgRNA of NC_003977.1 (subtype *ayw*) were identified by aligning anti-Cp RNA aptamers against the sequences of 16 strain variant HBV pgRNAs, chosen at random from the ~750 sequences then available in GenBank. Mfold[33] suggests that each of the matched sites is potentially able to fold into a stem-loop with an over-represented sequence, 5′-RGAG-3′, in the loop[12]. The three most highly conserved NC_003977.1 sites, PSs1–3, as oligonucleotides, trigger sequence/structure-specific in vitro assembly of Cp, mostly into $T = 4$ NCPs. Sequence alignment and Mfold[33] readily identifies putative PS homologues of these sites in the JQ707375.1 RNA sequence (Fig. 2; 'Methods'). Each of the JQ707375.1[34] PS homologues is predicted to fold into a stem-loop, the latter presenting a purine-rich tetra-nucleotide motif. PSs 1 & 3, have ideal consensus Cp-recognition motifs, -GGAG- & -AGAG-, respectively, whilst the homologous PS2, has a slight variation (-GAAG-). Their secondary structures and stabilities vary, as expected[12].

To investigate the PS-mediated NCP assembly hypothesis, and determine the roles of PSs in assembly in the context of the genome, we used a transcript assembled from the commercially-available strain JQ707375.1[34]. This was isolated from a patient

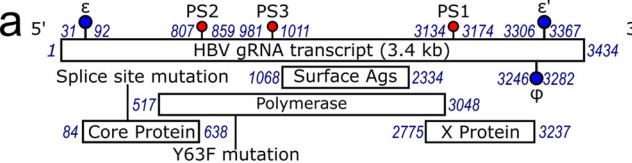

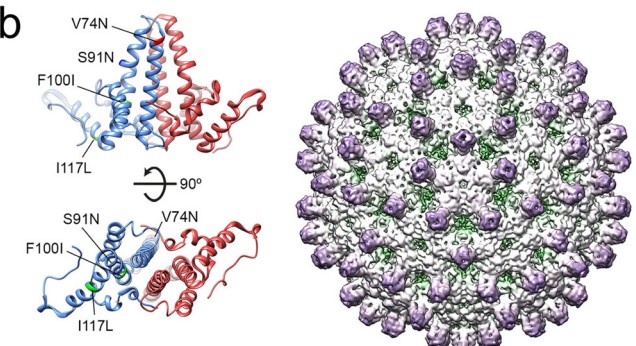

**Fig. 1 Locations of HBV RNA packaging signals & structure of the $T = 4$ NCP. a** Genetic map of the JQ707375.1 HBV pgRNA with the regions encoding open reading frames shown as bars below. The locations of its PSs, corresponding to homologous sites from NC_003977.1, are shown as lollipops, as are the ε and φ sites. Here and throughout, blue numbers refer to cognate genome nucleotide locations. **b** Left, Ribbon diagrams showing amino acids 1–143, of the Cp dimer, with the monomers coloured red and blue (PDB ID 3J2V) viewed from the side with the interior of the particle below (top) or from below (bottom). The locations of the amino acid changes between strains NC_003977.1 and JQ707375.1 are indicated on the blue monomers. Right, I1 reconstruction at ~4.7 Å resolution of the NC_003977.1 $T = 4$ HBV NCP (EMD-3715) reassembled around an oligonucleotide encompassing its PS1, viewed along a five-fold axis (bar = 100 Å)[12]. The projections correspond to the helices in the Cp dimer, left.

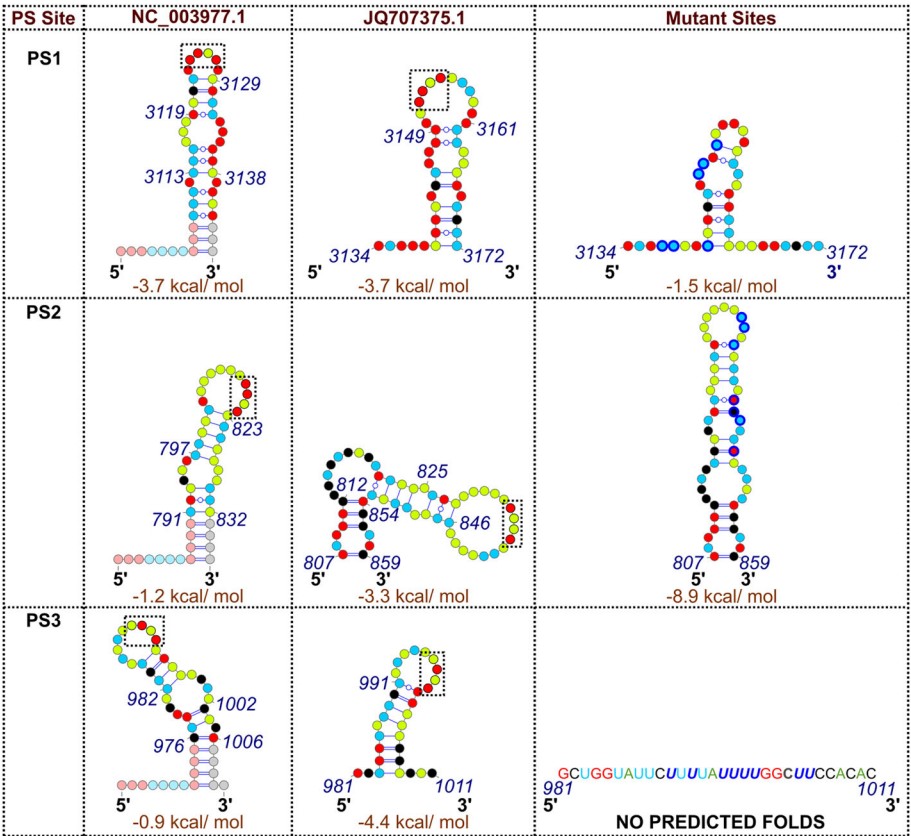

**Fig. 2 Homologous PSs in the two HBV strains and their variants.** HBV NC_003977.1 and JQ707375.1 PSs1-3, and their variant, sequences and Mfold secondary structures[12]. Faded nucleotides represent G:C clamps and 5′-leaders used to force the NC_003977.1 sequences to adopt single folds and to facilitate dye-labelling[12]. Here and throughout, RNA nucleotides are shown as coloured dots (green—A, black—C, red—G, blue—U), Watson–Crick base pairs are indicated as lines, which are interrupted by circles for G–U pairs. The predicted folding free energies of the unliganded RNA fragments are also shown. These do not include the non-HBV sequences. -RGAG- motifs are shown boxed (black), and mutated nucleotides within PS variants are outlined in dark blue.

with a lamivudine-resistant infection and is therefore thought to be replication competent. The ~3400 nt assembled RNA is closer to the size of full-length pgRNA but lacks the 5′ cap and 3′ polyadenylation. It shares 90.7% nucleotide sequence identity with the same region of NC_003977.1, from which the recombinant Cp used in assembly assays was expressed[12]. There are just 4 amino acid sequence changes between the Cp's of these strains: (NC_003977.1 vs. JQ707375.1, respectively) V74N; S91N; F100I & I117L, all of which lie within the first 149 amino acids of core (Fig. 1b) which form the outer portion of the NCP shell. None would be expected to interact with the pgRNA. NC_003977.1 Cp was therefore used in the in vitro assembly assays with the JQ707375.1 transcript.

**In vitro assembly with the JQ707375.1 RNA transcript.** PS-mediated assembly of ssRNA viruses at nanomolar concentrations[12,16,35] mimics the genome packaging specificity outcomes of natural infections[36]. These conditions differ from most in vitro reassembly studies which are usually carried out at much higher concentrations[37,38]. Previously, PS oligonucleotide-induced HBV NCP assembly was monitored using single-molecule (sm)fluorescence correlation spectroscopy (smFCS) with dye-labelled RNA oligonucleotides[12,16,39]. These assays monitor real-time changes to the hydrodynamic radius ($R_h$) of the labelled RNA, and end-products were analysed by negative stain EM (nsEM). However, the small sample sizes of such experiments prevent more detailed analysis. To overcome that limitation here, NCPs were reassembled in 96-well plates under

sm conditions using a liquid-handling robot (Fig. 3a). These samples were pooled and concentrated (~10 fold) to allow subsequent fractionation and further analysis. Well over 90% of the input RNA was recovered from this step, presumably with the rest lost in the concentrator. The subsequent analysis assumed that the material remaining soluble reflects the endpoint of each titration.

Titrations of Cp dimer, prepared from recombinant NCPs using 1.5 M guanidine hydrochloride (GuHCl), comprising 10 × 2 µL aliquots from one of six different stock concentrations, were made into an initial (180 µL) solution containing gRNA (1.1 nM). These dilute the gRNA to a final concentration of 1 nM, whilst spanning sub-stoichiometric to a 4-fold molar excess for formation of a $T = 4$ NCP around each gRNA. Each titration point was equilibrated for 10 min at room temperature (~20 °C) before addition of the next aliquot. This titration allows assembly initiation to occur at the highest affinity PSs on each gRNA molecule and then proceed to completion[37], and yields are >80% as judged by input gRNA. The final molar excess of Cp dimer ensures that any assembly defects detected are not a consequence of non-functional Cp dimers. Assembled NCPs appear as separated, negative-stain excluding particles in TEM, with $A_{260/280}$ ratios consistent with each particle containing a single, full-length copy of the transcript (see below).

The products from the titration reactions were analysed by absorption spectroscopy and the relative efficiency of complete NCP formation probed by treatment of one of half of the sample with RNase A. Both aliquots were then fractionated by size-

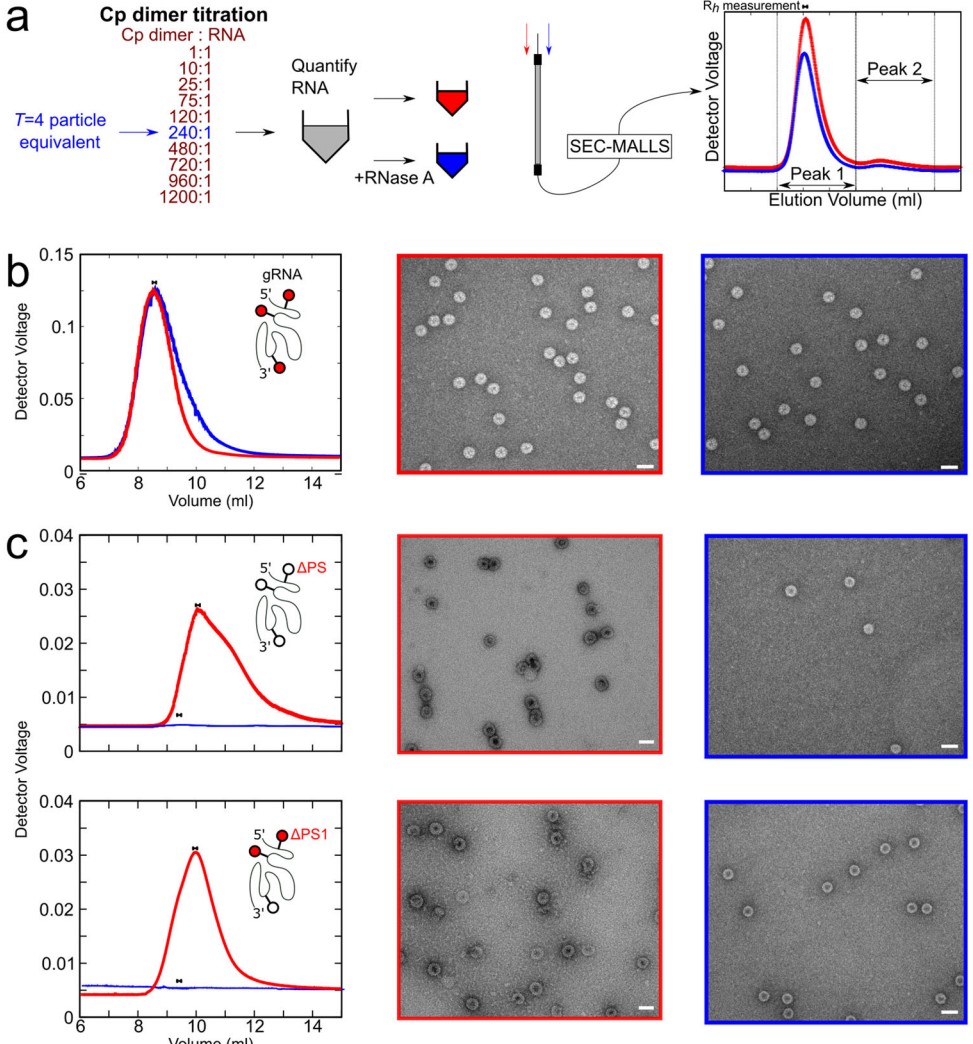

**Fig. 3 Roles of PSs in regulation of in vitro assembly of HBV NCPs. a** Cp dimer was titrated at 10 min intervals (left) into heat-annealed ('Methods') HBV gRNA transcript in a 96-well plate. Reactions were then pooled (grey) and split into 2 (red/blue), one of which was treated with RNase (blue). Aliquots of each were visualised by nsEM and the remaining sample analysed by SEC-MALLS chromatography, recording the elution volumes (graph, right) and hydrodynamic radii ($R_h$, bar above peak) of particulate reassembly products from their light-scattering ('Methods'). **b** Left: LS traces of the reassembly products with PS variants of 1 nM HBV gRNA and HBV Cp dimer: red and blue traces are pre- and post-RNase treatment, respectively. Middle/Right: nsEMs of the assembly products shown (left). Scale bars = 50 nm. **c** Left: LS traces, as in (**b**), with 1 nM ΔPS (top) and ΔPS1 (bottom), and Middle/Right: nsEMs of the assembly products shown. Data found in Supplementary Data 1.

exclusion chromatography with the eluting material detected via multi-angle, laser light-scattering (SEC-MALLS). Light-scattering peaks were collected, re-concentrated to ~1 mL, and their absorbance values re-measured. The SEC-MALLS signals were used to estimate assembly efficiency and the hydrodynamic radii ($R_h$) of eluting species (Fig. 3a, Table 1).

For JQ707375.1 RNA transcripts the reassembly products prior to RNase treatment (Fig. 3b) elute as a single, symmetrical peak, ~8.5 mL after application to the column. Unassembled Cp dimer, present from the titration, is much smaller than NCPs and invisible on these plots. RNase-treated aliquots elute essentially identically with a very similar yield, although their peak has a slight low-side shoulder. Prior to chromatography, both aliquots contain identical stain-excluding particles in nsEM, consistent with $T = 4$ NCPs. Their apparent $R_h$ values (~18–20 nm) match those for NCPs formed by Cp recombinant expression in *E. coli* (~19 nm, Supplementary Fig. 1), and are similar to values of ~25–32 nm determined by previous cryo-EM and smFCS

measurements, respectively. The results show that NC_003977.1 Cp dimer successfully reassembles nuclease-resistant $T = 4$ NCPs around JQ707375.1 gRNA (Table 1), validating the assumption about the common functionality of Cp's between HBV strains.

**Probing the role(s) of PSs1-3 in in vitro assembly of gRNA.** Sequence variants were designed to investigate the role(s) of PSs in regulating assembly with JQ707375.1 transcripts in the in vitro assembly assay. Variants remove loop recognition motifs and/or destabilize the secondary structures of the stems presenting the motif, whilst retaining wherever possible the global gRNA fold, as predicted by Mfold[33]. PS1 lies within an especially G-rich (50%) region and consequently encompasses several potential alternative -RGAG- motifs. This made subtle motif ablation impossible. Instead, six G to U mutations were introduced into the 5′ leg of the PS stem destabilising it (predicted free energy of folding going from −3.7 to −1.3 kcal mol$^{-1}$), and increasing the bulge

**Table 1 Characterisation of NCPs reassembled around gRNA constructs.**

| gRNA | $A_{260}$ (Before and after RNase A addition) | $A_{260/280}$ | RNA yield (%) | $R_h$/nm |
|------|------|------|------|------|
| gRNA | $0.12 \rightarrow 0.097$ | 1.49 | 81 | 18.2 |
| ΔPS | $0.019 \rightarrow 0.0067$ | 0.91 | 35 | 6.2 |
| ΔPS1 | $0.048 \rightarrow 0.004$ | 1.07 | 8 | 9.2 |
| ΔPS2 | $0.128 \rightarrow 0.095$ | 1.55 | 74 | 18.4 |
| ΔPS3 | $0.119 \rightarrow 0.069$ | 1.3 | 58 | 17.2 |

All values are post-nuclease treatment unless stated.

size, whilst moving the -RGAG- motif to a more central position within the loop (Fig. 2). For PS2 and PS3, purine to U mutations can be introduced directly into the loops, replacing their recognition motifs -GAAG- and -AGAG-, with -UUAU- and -UUUU-, respectively. It was also possible to destabilize the base-pairing in their stems via the following substitutions: in the 3′ leg of the PS2 stem-loop (-UUAAAAUUA- to -UAGCUUUG- (nts 841–848)), and either side of the loop in PS3 (-AUAUAUUUUGGGAA- to -UUUUAUUUUGGCUU- (nts 992–1005)). These substitutions prevent formation of alternate -RGAG- motifs in either PS. Disrupted PS2 is predicted to fold as a very stable ($-8.9$ kcal mol$^{-1}$) stem-loop, with a multiply interrupted stem. The mutations within PS3 ablate all possible secondary structures using the default Mfold parameters (Fig. 2). The global folding effects of these changes were estimated by folding a 300 nt region centred on each PS using a sliding 60 nt window noting the frequency of defined structures. This procedure identified no long-range base-pairing issues and confirmed the greatly reduced occurrence of a Cp-recognition motif in a loop in the PS1 variant, together with the complete ablation of PS2 and PS3 recognition signals and secondary structures.

The effects of the variant PSs on NCP assembly were assessed in transcripts containing all three variant PSs (ΔPS), or with individual variants (ΔPS1, ΔPS2 and ΔPS3) (Fig. 3c, Table 1). ΔPS gRNA does not assemble $T = 4$ NCPs significantly under these conditions. Its pre-RNase assembly product elutes later than a bona fide NCP (10 mLs after being applied to the column) and has a pronounced low-side tail. In nsEM, it appears to produce small quantities of malformed and aggregated particles which are readily penetrated by stain. Absorbance measurements are consistent with this, the peak containing only ~10% of the wild-type amount of RNA. Particle formation, however, remains RNA-dependent since RNase treatment eliminates all light-scattering material, revealing only a small number of apparently correctly assembled particles. This result confirms the assumption that these sites act as PSs within JQ707375.1 gRNA. Alteration of only 24/3200 nucleotides in this RNA transcript completely prevents assembly. Since it is the same length as the wild-type sequence, assembly cannot rely purely on favourable electrostatic interactions, as has been proposed for many single-stranded (ss) RNA viruses[40–45], i.e. it implies assembly regulation by the RNA. The sequence variations in ΔPS gRNA do not alter its hydrodynamic radius significantly (Supplementary Fig. 2) confirming that this regulation is based on sequence-specific recognition of PS sites.

ΔPS1 pgRNA also assembles poorly compared to the wild-type (Fig. 3c, Table 1), eluting (~9.5 mL vs ~8.5 mL for WT) as a roughly symmetrical peak. The material before RNase treatment consists mostly of separate but misshapen particles of widely differing radii, which are mostly freely penetrated by negative stain. As with the ΔPS variant, assembly is dependent on the RNA, with RNase treatment eliminating all light-scattering

material, although there are many more particles roughly the size of an NCP than with the ΔPS variant transcript. They are all freely penetrated by negative stain. These results arise from variation of just 6 nucleotides across the gRNA, and an -RGAG-motif is still present in the smaller loop of the variant. Since the ΔPS and ΔPS1 transcripts have distinct assembly properties, PSs 2 & 3 must also contribute differentially to assembly regulation. Analysis of their individual reassembly reactions suggests, however, that these are less significant than for PS1 (Supplementary Fig. 3). Both variants produce products that co-elute with bona fide NCPs assembled around wild-type gRNA. They elute as symmetrical peaks containing mostly stain-excluding, single particles. Their hydrodynamic radii are also very close to that of wild-type (~17–18 nm vs ~18 nm), although their susceptibility to RNase (Table 1) implies that they fail to form completely closed shells. ΔPS3 appears more deleterious in terms of closed shell formation than ΔPS2.

These results are consistent with a PS-mediated in vitro assembly mechanism for the HBV JQ707375.1 gRNA. Multiple PSs within the genomes of ssRNA viruses[15,20,46] vary around a consensus sequence defining a hierarchy of binding affinities for cognate capsid proteins. This preferred assembly pathway around cognate viral RNA prevents multiple assembly initiation events occurring on the same RNA, ensuring efficient capsid assembly, and avoiding the formation of off-pathway kinetic traps[37]. It implies that PSs act cooperatively but differentially during assembly, as seen here. This mechanism creates a fixed spatial relationship between the pgRNA and Cp shell, and for several RNA bacteriophages these interactions have been revealed directly by asymmetric cryo-EM reconstruction[47,48]. Zlotnick and colleagues[49] have proposed that a preferred pgRNA conformation within the NCP would facilitate its reverse transcription. To investigate that possibility, we determined the structure of the NCP assembled around the wild-type RNA transcript.

**The gRNA containing NCP has ordered internal density**. A cryo-EM reconstruction was calculated using particles of purified NCPs containing the HBV JQ707375.1 RNA transcript ('Methods' & Table 2). After data collection, 127,410 $T = 4$ (84%) and 23,257 $T = 3$ (16%) particles were chosen for further analysis. The ratio of $T = 4:T = 3$ particles in this dataset is similar to that obtained following reassembly around the NC_003977.1 PS1 oligonucleotide[12]. 2D classification reveals that most $T = 3$ particles and ~18% of the $T = 4$ particles were heterogeneous, with many particles appearing empty. No further image processing was performed on these. A homogenous dataset of the remaining $T = 4$ particles was subjected to icosahedrally-averaged refinement yielding a ~3.2 Å resolution map. This reconstruction reveals a Cp layer similar to those seen in previous EM and crystal structures. Density for Cp chains is complete up to the start of the ARD (Fig. 4, Supplementary Fig. 4)[50–53]. The four Cp monomers of the asymmetric unit were built into the map using a previous structure of the NC_003977.1 Cp[53] (PDB 3J2V) as a starting model. At this resolution, the four distinct Cp monomers of the quasi-equivalent capsomer, chains A-D, can be built using polypeptide chains 144, 143, 144 and 147 residues long, respectively (Supplementary Figs. 4 and 5). The icosahedrally-ordered Cp surface lattice surrounds a second, less-ordered internal layer of density, which may well include density for the encapsidated gRNA transcript and the Cp ARD domains.

Reconstruction of the same dataset without the imposition of symmetry fails to reveal any further molecular detail at 4.2 Å resolution, the internal density appearing as a mostly continuous layer underneath the Cp shell (Supplementary Fig. 6). To reduce

**Table 2 Data for Cryo-EM structures.**

|  | HBV NCP sym I1 EMD-11700/PDB: 7ABL | HBV NCP sym exp EMD-11702 | HBV NCP sym C1 EMD-11701 |
|---|---|---|---|
| *Data collection and processing* |  |  |  |
| Microscope | FEI Titan Krios | FEI Titan Krios | FEI Titan Krios |
| Detector | FalconIII (linear mode) | FalconIII (linear mode) | FalconIII (linear mode) |
| Magnification | ×75,000 | ×75,000 | ×75,000 |
| Voltage (kV) | 300 | 300 | 300 |
| Electron exposure (e$^-$/Å$^2$) | 53.1 | 53.1 | 53.1 |
| Exposure per frame (e$^-$/Å$^2$) | 0.9 | 0.9 | 0.9 |
| Defocus range (μm) | −0.75 to −2.5 | −0.75 to −2.5 | −0.75 to −2.5 |
| Pixel size (Å) | 1.065 | 1.065 | 1.065 |
| Micrographs collected (no.) | 2658 | 2658 | 2658 |
| Initial particles (no.) | 150,667 | 104,337*60 = 6,260,220 | 150,667 |
| Final particles (no.) | 104,337 | 1,254,013 | 104,337 |
| Symmetry imposed | I1 | C1 | C1 |
| Map resolution (Å) | 3.2 | 3.5 | 4.2 |
| FSC threshold | 0.143 | 0.143 | 0.143 |
| Map resolution range (Å) | 3.02–3.59 | 3.3–6.7 | 3.7–7.2 |
| *Refinement* |  |  |  |
| Model resolution (Å) | 3.21 |  |  |
| FSC threshold | 0.5 |  |  |
| Mask correlation coefficient | 0.87 |  |  |
| Map sharpening B factor (Å$^2$) | −158 |  |  |
| Model composition |  |  |  |
| Non-hydrogen atoms | 277,380 |  |  |
| Protein residues | 34,680 |  |  |
| ADP (B-factors) min/max/mean |  |  |  |
| Protein | 9.60/85.84/27.30 |  |  |
| R.m.s. deviations |  |  |  |
| Bond lengths (Å) | 0.007 |  |  |
| Bond angles (°) | 0.857 |  |  |
| Validation |  |  |  |
| MolProbity score | 1.30 |  |  |
| Clashscore | 2.57 |  |  |
| Rotamer outliers (%) | 0.58 |  |  |
| Ramachandran plot |  |  |  |
| Favored (%) | 96.14 |  |  |
| Allowed (%) | 3.68 |  |  |
| Outliers (%) | 0.18 |  |  |

Cryo-EM data collection, refinement and validation statistics.

the impacts of the Cp layer on the reconstruction, the signal from the globular region of the Cp was subtracted from the particles in the same dataset, and a 3D classification by alignment of the remaining density calculated revealing five similar classes (not shown). The icosahedrally-averaged reconstruction of the $T = 4$ particle was then symmetry expanded[21–24] following removal of much of the Cp shell using a spherical mask. The remaining density was subjected to further 3D classification without alignment into a further five roughly equally populated classes (Supplementary Fig. 7a). These were reconstructed without imposing symmetry[21,54], yielding maps with global resolutions of 3.5–3.6 Å (Supplementary Fig. 7). The maps were low-pass filtered to 5 Å resolution for clarity revealing an asymmetric cage-like density under the Cp shell at this lower resolution, presumably corresponding to either the C-terminal ARDs or gRNA, or both, in close contact with the Cp shell (Fig. 4, Supplementary Fig. 7). Superposition of the internal density obtained by asymmetric reconstruction of the NCP assembled around NC_003977.1 PS1 (EMD-3714), into this map shows that this feature, which we presume includes multiple PS RNA oligonucleotides[12], overlaps with the internal density seen with the NCP gRNA. Additional refinement failed to resolve these features further here, perhaps due to conformational heterogeneity or flexibility[55,56].

Interestingly, these maps reveal multiple fingers of density at every particle vertex that transit the gap between the Cp layer and the internal density. Such bridges, illustrated in Fig. 4 based on the Class 3 data (Supplementary Fig. 7d) could be the result of the formation of multiple Cp-gRNA contacts in the NCP, i.e. as expected for PS-mediated assembly. These density bridges 5-fold and 2-fold positions (Fig. 4c, d and g, h, respectively) appear to be different at the different symmetry axes. For both, the internal density fuses with that of the Cp subunits adjacent to residue Pro144, i.e. just before the start of the ARD domain, and could represent extensions of the polypeptide chain at these points. The bridging contacts at the 3-fold (Fig. 4a, b) and quasi-3-fold vertices (Fig. 4e, f) are much more similar to each other but distinct from those at 2- and 5-fold axes. They fuse into the density of the Cp around residues E40 and C48. Both these amino acids are in well-ordered sections of the globular fold and there is no unassigned Cp density in this part of the map.

In order to test the idea that there are RNA PS-Cp contacts in the reassembled NCP, we used XRF[27,57,58]. To see if PS1, the most important PS within the JQ707375.1 transcript for regulating assembly in vitro (Fig. 3c) contacts Cp in the reassembled NCP, XRF nucleotide reactivities across a genomic region encompassing PS1 were determined for both NCP and free transcript ('Methods'). Both reassembled NCP and transcript

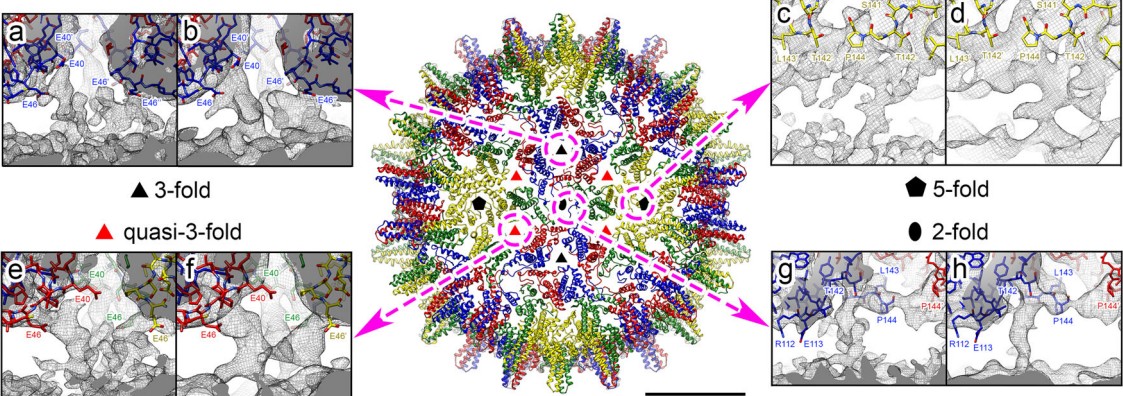

**Fig. 4 Cryo-EM reconstruction of the *T* = 4 NCP formed with JQ707375.1 gRNA.** Centre: Atomic model of the front-half of the HBV *T* = 4 NCP shown as ribbon diagrams, coloured in yellow (A monomer), green (B monomer), blue (C monomer) and red (D monomer) for the quasi-equivalent monomers, and viewed along a two-fold axis built into the icosahedrally-averaged cryo-EM density map at 3.2 Å resolution (bar = 100 Å). Symbols indicate icosahedral symmetry axes. **a–h** Dashed (pink) circles indicate the locations of individual NCP symmetry axes, including the quasi-3-fold, and the arrows lead to boxes showing the transiting internal density associated with each of these locations. Cryo-EM density maps are shown as a grey mesh with the centre of the NCP towards the bottom. Cp atomic models are shown as sticks. Amino acid side-chains adjacent to the internal layer of density are indicated and coloured by heteroatom. The atomic model is fitted into the non-filtered shell of the Class 3 density map obtained after symmetry expansion and focused classification (**a, c, e, g**), and for clarity into the same map low-pass filtered to 5 Å (**b, d, f, h**), both shown at 1.5 σ. **a, b** Contacts located at three-fold vertices involve the E40-C48 sequence in Cp (including the E40-C43 alpha-helix and the S44-C48 loop) of three C monomers. The side chains of residues E40 and E46 are directly pointing to the internal density shell. **c, d** Contacts located at five-fold vertices involve five A monomers (pentamer). **e, f** Contacts located at quasi-three-fold vertices involve the E40-C48 sequence in Cp (including the E40-C43 alpha-helix and the S44-C48 loop) of one A monomer, one B monomer and one D monomer. The side chains of residues E40 and E46 point directly at the internal density shell. **g, h** Contacts located at two-fold vertices involve two C monomers and two D monomers (hexamer).

were flash-frozen and exposed to the X-ray beam on the customised beam-line at the NSLS-II in Brookhaven for various times ranging from 0 to 100 milliseconds[59]. Samples were returned to the host laboratory in the frozen state and RNA extracted/prepared for primer extension with a dye-labelled DNA primer annealed at nucleotide positions 3308–3328 (Fig. 5). The reverse transcription extension products were then analysed using capillary electrophoresis (CE), and a combination of published and in-house software[27,58]. Reassembled and protein-free gRNA is predicted to adopt the same stem-loop structure at PS1. This fold places the -RGAG- recognition motif (boxed in Fig. 5c) in the 3′ half of the loop, as expected (Fig. 2). The reactivities of the nucleotides in these two RNA stem-loops are shown colour coded (dark green to red corresponding to unreactive to highly reactive, respectively), and as a difference map in Fig. 5d. Nucleotides within the loop show the most significant differences in reactivity between the two states. In the transcript, the nucleotides of the -RGAG- motif have high reactivities (red), with the 5′ and 3′ neighbouring nucleotides being much less reactive. In the NCP, however, these reactivity patterns across the motif are largely reversed. All four nucleotides of the recognition motif become largely unreactive, whereas their 5′ and 3′ neighbours become more reactive, an effect that extends into the uppermost base pair of the stem (Fig. 5b, c). These data are consistent with a direct interaction between the NCP Cp layer and the PS1 recognition motif. There are also slight conformational rearrangements in the loop in the NCP resulting in increased flexibility of the non-contacted nucleotides.

In vivo NCP assembly is known to be affected by Cp phosphorylation and/or the binding of replicase at the ε site[28–32], both of which are missing in the in vitro assay. The JQ707375.1 transcript encompass both ε and φ sites (Fig. 1), and these sites are thought to form long-range base pairs making the RNA topology important for assembly[60,61]. In order to test this idea, we carried out reassembly with a variant RNA transcript of its ε site (Supplementary Fig. 2) preventing it from base-pairing with the φ site.

Strikingly, this variant gRNA, unlike those carrying the PS mutations, has a significantly increased hydrodynamic radius (27 ± 0.4 vs. 17 ± 0.3 nm, respectively). It is also a poor in vitro assembly substrate, although not as poor as the ΔPS and ΔPS1 variants, implying that in vitro and presumably also in vivo, both RNA topology and PS-Cp contacts contribute to assembly efficiency. It has been argued that ε is the most important pgRNA sequence since in vivo it is possible to encapsidate heterologous RNAs apparently encompassing only this site from the HBV genome. However, in these experiments the ε site was flanked by additional HBV genomic sequences which sequence/S-fold analysis (Supplementary Methods) suggests contain multiple unrecognised PS sites akin to those analysed here. In addition, in a fusion of this fragment to a 3′ *lacZ*, an assumed non-specific heterologous RNA construct, an additional PS site is added. The combinatorial effects of such sites would explain their packaging, ε sites being packaged whilst *lacZ* alone is not (Supplementary Tables 1–3).

## Discussion

The assembly of infectious HBV is a complex process that many people have studied in vivo in suitable cell culture systems. Here we have used a minimised molecular system to investigate whether the RNA stem-loops identified previously stimulate in vitro assembly of an RNA transcript encompassing most of the coding region of a strain not previously part of our analysis (JQ707375.1). The results establish that in such an assay this gRNA transcript is efficiently encapsidated. Modification of the gRNA to prevent or alter presentation of the Cp-recognition sequence (-RGAG-), at three putative PS homologues of sites studied previously as oligonucleotides, results in varying assembly deficits. The -RGAG- Cp recognition motif of one such site, PS1, appears to be selectively protected from X-ray modification in an HBV NCP, compared to a free RNA transcript. These results are consistent with NCP formation in vitro occurring via a PS-mediated assembly mechanism. Given that the same RNA and Cp

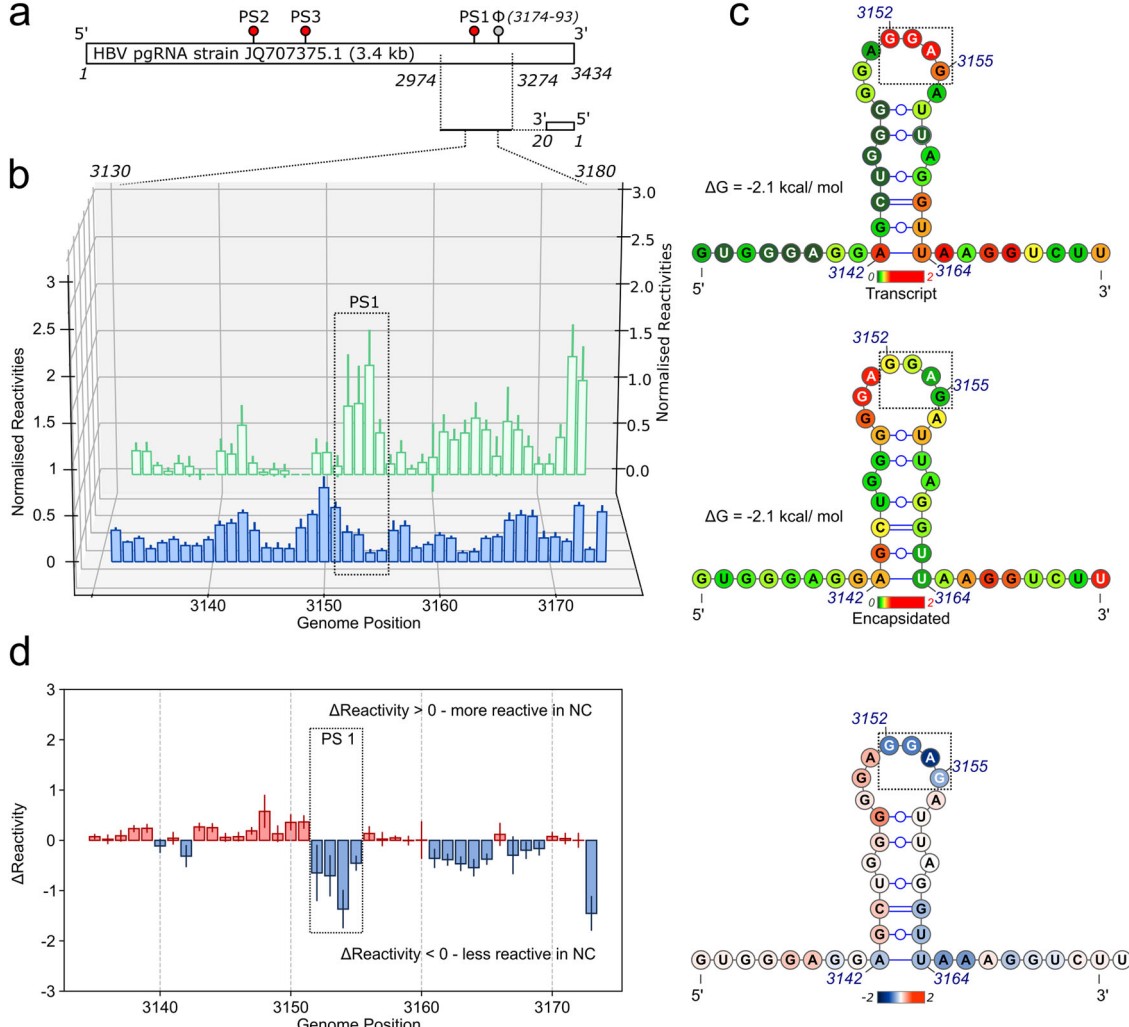

**Fig. 5 RNA footprints of the PS1 region. a** Protein-free transcripts of the gRNA, or following reassembly and purification of NCPs, were hydroxyl radical footprinted as frozen samples by exposure to X-rays[25,26,57]. RNAs were recovered from frozen samples. Reverse transcription from a 5′fluorescently-end-labelled primer complementary to the pgRNA 3′-end was used to assess nucleotide reactivities in a fragment encompassing PS1 using capillary electrophoresis. **b** Waterfall plot of the per nucleotide reactivities and associated standard errors from triplicate samples of transcript (green) or NCP (blue). Data found in Supplementary Data 2. **c** Secondary structures of the PS1 folds showing their XRF reactivities. Nucleotides are coloured red to green depending on their relative reactivity on exposure to the beamline (red → green = reactive → inert (reactivity of 2.0 → 0)), key shown below each fold. **d** Left: The average reactivity at each nucleotide across the PS1 sequence is shown. Red/blue indicate positions more/less reactive in the NCP. Right: Average reactivity at each nucleotide shown upon the secondary structure of PS1. Nucleotides are coloured blue/red depending on their reactivity within the NCP (blue → red = less → more reactive (Δreactivity of −2.0 → 2.0)), key shown below the PS1 fold. Data found in Supplementary Data 2.

sequences will participate in NCP formation in vivo, it seems reasonable to conclude that a similar mechanism contributes to such assembly.

Cryo-EM reconstruction of the JQ707375.1 RNA transcript assembled NCP, reveals extensive density below the globular folds of the Cp shell. Symmetry expansion suggests that there are multiple contacts between this internal density and the outer shell at particle symmetry axes. It is not possible at the resolutions obtained to assign this internal density to either the C-terminal Cp ARD domains, the gRNA transcript or a complex of both. Since this density is not present in NCPs lacking encapsidated RNA[12], it is clearly a consequence of Cp-gRNA interaction. Previously, we showed that individual oligonucleotides encompassing PS sites will trigger sequence-specific NCP formation in vitro. Asymmetric reconstruction of the NCP revealed an inner density larger than could be accounted for by a single RNA stem-loop[12], although we assumed that PS1, or multiple copies of it, were part of this density. Superposition of that density on the

inner layer seen with the gRNA here shows that they are coincident with respect to radial positions, suggesting that this layer does contain RNA. XRF of PS1 confirms this assumption. PSs only need to function, i.e. contact the protein shell of a virion, during assembly. For bacteriophage MS2[47], many of its RNA PSs dissociate from the CP post-assembly. The fact that PS1 remains in contact after NCP assembly may reflect the need in a para-retrovirus to create an 'RNA track' along which the polymerase must move within the NCP as it reverse transcribes the gRNA[12,49].

The in vitro assembly data with the three variant PS sites suggest that they constitute a novel, evolutionarily stable aspect to NCP assembly regulation. The experiment with the ε variant suggests that gRNA topology is also important in efficient assembly, i.e. that there may be multiple factors that assist the process in vivo. The PS-mediated aspect of assembly is a direct anti-viral drug target that could be exploited in the global challenge to treat chronic HBV infections[62]. There are currently only

a limited set of clinical interventions, including polymerase inhibitors Lamivudine and Tenofovir, and interferon[63–65]. Recently novel ligands that target additional aspects of the viral lifecycle have been identified[66–70]. Amongst these are the 'capsid assembly modulators, CAMs' that include the hetero-aryl, dihydropyrimidine (HAP) compounds characterised by Zlotnick and colleagues[71,72]. They have shown that HAPs bind Cp dimers altering their assembly behaviour resulting in formation of 'empty' NCP particles. Mathematical modelling suggests that targeting capsid assembly via distinct routes would be synergic[73,74]. The recent development of a mouse model in which to probe the HBV lifecycle[75] will hopefully allow these ideas to be tested rapidly allowing the Global Challenge presented by this virus to be met.

## Methods

**Expression and purification of HBV Cp.** HBV Cp was expressed in *E. coli* BL21(DE3) cells (T7 Expression strain, New England Biolabs), expressed from a pET28b plasmid. Induction with 1 mM isopropyl-β-D-thiogalactoside (IPTG) at an optical density (OD$_{600}$) of ~0.6 was followed by growth for 20 h at 21 °C. Cells were lysed using a Soniprep 150 and clarified by spinning at $11,000 \times g$ for 1 h. NCPs were then pelleted by centrifugation at $120,000 \times g$ for 14 h. Pellets were resuspended in 20 mM HEPES (pH 7.5), 250 mM NaCl, 5 mM dithiothreitol (DTT) and applied to a prepacked Captocore 700 column (GE Life Sciences). Fractions containing NCPs were pooled and precipitated with 40% (w/v) ammonium sulfate. NCPs were dissociated into Cp dimers by dialysis into 1.5 M guanidinium chloride (GuHCl) as previously described[12,38]. All steps after sonication were performed in the presence of complete protease inhibitor tablets (1 tablet per 500 mL buffer, Thermofisher Scientific). Cp dimer concentration was determined by the absorbance at 280 nm (ε$_{280}$ of Cp$_2$ = 55,920 L mol$^{-1}$ cm$^{-1}$). Cp fractions with an A$_{260/280}$ ratio of ≤0.65 were used in assembly assays.

**Identification of homologous PS sites in strain JQ707375.1.** PS sites within the JQ707375.1, homologous to those found within NC_003977.1, were identified by analysis of stem-loop folds within a 300 nt region of the genome that aligns to the equivalent regions encompassing each PS in the latter strain. A 60 nt reference frame was slid across that region and secondary structure of the corresponding RNA sequence sampled, recording each unique secondary structure fold with negative free energy[12]. Previously such PS sites were numbered according to the most frequently matched sites in a Bernoulli Plot of the aptamer sequences against the cognate pgRNA[12]. We retain those labels for the homologous JQ707375.1 strain sites, although this means their numbering does not reflect their relative order. In JQ707375.1, the PS1 equivalent lies between nucleotides 3134–3172 (Fig. 2). Mfold suggests that it retains the bulge in the base-paired stem and the -RGAG- loop motif seen in NC_003977.1, although the latter is at the 5′ side of the loop as opposed to being central. This PS1 lies in the gene encoding the X protein, and its nucleotide sequence creates two non-synonymous codons encoding conservative amino acid changes. PS2 (nts 807–859) and PS3 (nts 981–1011) sit in regions of the polymerase gene with much lower sequence identities than the overall genome (67 & 75%, respectively). PS2 has a slightly modified loop motif (-RAAG-), which is centrally located, as opposed to being at the 3′ side of the loop, as it is in the NC_003977.1 strain. The loop sits on top of a three base-paired duplex leading to single-stranded bulges of differing sizes. The upper region of this PS stem-loop is homologous to the PS2 oligonucleotide fold in NC_003977.1 (Fig. 2). The JQ707375.1 PS3 retains the 3′ positioning of the -RGAG- motif within the loop but has a more extensively base-paired stem than in the previous strain (Fig. 2). Nucleotide changes in JQ707375.1 PS2 create one conservative (F to Y), and two semi-conservative (T to N & H to L) amino acid substitutions relative to the NC_003977.1 polymerase. The sequence of the PS3 region yields only synonymous coding changes.

**Preparation of pregenomic RNA.** A wild-type gRNA transcript clone was assembled using pAM6 39630™, strain acc. no JQ707375.1, purchased from ATCC®. The RNA sequence was copied from the purchased cDNA in fragments using PCR, and the fragments cloned into the correct order between the *BspHI* and *HindIII* sites in a pACYC184 vector, using a Gibson Assembly® Master Mix, according to the manufacturer's protocol (New England Biolabs). The sequence of this construct was confirmed by Sanger sequencing (Source Bioscience, Nottingham). HBV pgRNA constructs encompassing PS mutations were produced synthetically, using gene fragments purchased from IDT. Gene fragments were cloned between the *BglII* and *HindIII* sites within an empty pET22b vector using a NEBuilder® HiFi Master Mix, according to the manufacturer's protocol (New England Biolabs). All pgRNA constructs were designed with a T7 promoter sequence at the 5′ end. Transcription of pgRNA constructs were carried out using a Hiscribe™ T7 High Yield RNA Synthesis Kit (New England Biolabs), after linearization of the DNA plasmid using *HindIII*. RNA was annealed prior to each

experiment by heating to 70 °C for 90 secs and cooling slowly to 4 °C in a buffer containing 50 mM NaCl, 10 mM HEPES and 1 mM DTT at pH 7. Products were assessed using a 1% (v/v) denaturing formaldehyde agarose RNA gel. RNA concentration was determined using the A$_{260}$ value (ε$_{260}$ of pgRNA = 32,249,500 L mol$^{-1}$ cm$^{-1}$).

**HBV NCP Assembly assays.** 180 μL of 1.1 nM annealed (as above) gRNA transcript in a buffer containing 20 mM HEPES (pH 7.5), 250 mM NaCl and 5 mM DTT, was incubated in each well of a 96-well plate at room temperature for 30 min. Cp dimer in dissociation buffer (as above) was then titrated into the RNA using a Biomek 4000 liquid handling robot (Beckmann Coulter), step-wise up to a ratio of 1200:1 Cp dimer: RNA. Ten 2 μL titrations were performed, utilising six different stock Cp dimer solutions (indicated in brackets): cumulatively adding 1 nM (100 nM), 10 nM (1 μM), 25 nM (2.5 μM), 75 nM (7.5 μM), 120 nM (12 μM), 240 nM (12 μM), 480 nM (24 μM), 720 nM (24 μM), 960 nM (24 μM) and 1200 nM (24 μM Cp dimer. This results in a final well volume of 200 μL, a cumulative volume of 19.2 mL/plate, and final concentrations of RNA and Cp of 1 nM and 1.2 μM, respectively. Cp aliquots were calculated to always reach 10% of the total reaction volume limiting the final concentration of GuHCl to 0.15 M in each aliquot. Following incubation at room temperature for 1 hr, the samples were pooled and concentrated to a final volume of 2 mL. The A$_{260/280}$ ratio was measured using a Nanodrop™ One (Thermofisher Scientific) and RNA concentration was calculated using absorbance at 260 nm. This sample was split into 2, with one half treated with 1 μM RNase A, and incubated overnight at 4 °C. After incubation, 5 μL of each sample was visualised by nsEM to assess particle shape and intactness, and the remaining sample analysed by application to a TSK G6000 PWXL column (Tosoh), in a buffer containing 20 mM HEPES (pH 7.5), 250 mM NaCl and 5 mM DTT, using a SEC-MALLS system (ÄKTA Pure (GE Heathcare) connected to a Optilab T-REX refractometer and miniDAWN® multiple angle laser light-scatterer fitted with a Wyatt QELS DLS module (Wyatt Technology)). Light scattering peaks were collected and concentrated to ~1 ml, where the RNA content of particles and yield considering the RNA input was once again determined using the absorbance at 260 nm. Absorbance values at 260 and 280 were corrected for light-scattering throughout, using the absorbance values at 310 and 340 nm as previously described[76]. All assembly assays were performed in triplicate.

**R$_h$ determination of gRNA transcripts.** 500 μL of gRNA transcript at a concentration of 500 ng/μL was applied to a TSK G6000 PWXL column (Tosoh) using the buffer and SEC-MALLS system described above. All measurements were performed in triplicate.

The hydrodynamic radius of assembled NCPs and gRNA transcripts were determined at the apex of peaks eluted from a TSK G6000 PWXL column. The light scattering at this point is quantified via the second order correlation function, (1) where $I(t)$ is the intensity of scattered light at time $t$.

$$g^2(\tau) = \frac{(I(t)I(t+\tau))}{(I(t)^2)} \qquad (1)$$

The measured correlation function is fit to Eq. (2) using a nonlinear least squares fitting algorithm to calculate the decay rate $g$.

$$g^2(\tau) = B + \beta \exp(-2\Gamma\tau) \qquad (2)$$

$G$ is then converted to the diffusion constant $D$ using Eq. (3), where $q$ is the magnitude of the scattering vector given by Eq. (4).

$$D = \frac{\Gamma}{q^2} \qquad (3)$$

$$q = \frac{4\pi n_0}{\lambda} \sin\left(\frac{\theta}{2}\right) \qquad (4)$$

$D$ is then fit to the Stokes-Einstein Eq. (5) to give the R$_h$.

$$R_h = \frac{kT}{6\pi\eta D_t} \qquad (5)$$

**X-ray footprinting of gRNA-containing NCPs.** To detect Cp-PS1 interactions in the NCP we used XRF, comparing the footprint on the encapsidated gRNA with the transcript used in the assembly reaction. Both samples were flash frozen and exposed to synchrotron X-rays at the 17-BM beamline (XFP) of the NSLS-II at the Brookhaven National Laboratory ('Methods')[26]. X-ray photons are largely absorbed by the solvent water which photolyzes to create hydroxyl radicals that then modify the ribose sugars of solvent accessible nucleotides in a flexibility-dependent manner[25,26]. Ribose sugars of single-stranded nucleotides are more reactive than those in base-pairs or involved in other molecular contacts. Modification leads to cleavage of the phosphodiester bond[57], and the frequency of cleavage at each nucleotide can be determined by reverse transcription using fluorescently-tagged primers annealed 3′ to the site of interest.

NCPs were re-assembled as above around the wild-type JQ707375.1 pgRNA in three 96-well plates. These samples were concentrated by centrifugation, and purified over a 10–50% (w/v) sucrose density gradient using a SW 40 *Ti* rotor at $190,000 \times g$ for 2 h. Purified NCP samples were diluted in 10 mM sodium

phosphate buffer (pH 7.4) to 200 ng/μL with respect to pgRNA using a Nanodrop™ One, and flash-frozen using $LN_2$ as 5 μL aliquots in 8 tube PCR strips. Heat-annealed (see above) pgRNA only samples at the same concentration were also flash-frozen as a control.

The samples were footprinted on beamline 17-BM (XFP) at the National Synchrotron Light Source II (Brookhaven National Laboratory, NY, USA). A calibration curve of the X-ray induced photo-bleaching of Alexa488 fluorophore, diluted in 10 mM sodium phosphate buffer (pH 7.4) was performed to ascertain beam strength, allowing adjustment of sample exposure times between runs to keep levels of RNA cleavage events similar between experiments. Samples were mounted in the beamline in a temperature-controlled (−30 °C, ensuring samples remained frozen), 96-well motorised-holder, which accommodates strips of 8–12 PCR tubes. Beam exposure was controlled using a Uniblitz XRS6 fast shutter (Vincent Associates), exposing samples for 10, 25 and 50 msec, with each time point performed in triplicate. Ideally, exposed samples contained no more than 1 cleavage event across the region of interest, defined as the length of the reverse transcript. This assumes that cleavages elsewhere do not cause large-scale conformational changes in the frozen gRNA within the time span of the exposure (max = 100 milliseconds). Similar data on bacteriophage MS2 gRNA yield predicted secondary structures similar to those seen at atomic resolution by cryoEM[47,58], confirming that this is a reasonable assumption.

**Analysis of X-ray induced RNA modification**. RNA was extracted from exposed NCPs using the phenol-chloroform (Thermofisher Scientific) extraction technique according to the manufacturers' protocols, with the exception that the RNAs were precipitated overnight at −20 °C in the presence of 1 volume isopropanol, 0.3 M sodium actetate (pH 5.2) and RNA-grade glycogen (0.01×, Thermofisher Scientific) which gave us higher yields. Recovered RNA was washed 3× with 70%(v/v) ethanol and allowed to air dry for 5 min before resuspension with 12 μL nuclease-free water. The beamline-exposed extracted and free pgRNA were reverse transcribed using Superscript IV (Thermofisher Scientific) and a sequence specific 5′ 5(6)-carboxyfluorescein (FAM) labelled primer that attached 3′ of the region of interest. Sequencing ladders were synthesised from in vitro transcribed RNA using Hexachloro-fluorescein (HEX) or 5-carboxytetramethylrhodamine (TAMRA) labelled primers and the addition of a 3:1 molar concentration of ddATP or ddCTP, respectively. Bound RNA was degraded with 5 units of RNase H (New England Biolabs) and the cDNA purified by ethanol precipitation overnight at −20 °C (3× volumes of ethanol, 0.3 M sodium acetate 0.01× volume of glycogen). Experimental and sequencing ladder cDNAs were then resuspended in 20 μL formamide, their concentration measured by $A_{260}$ absorbance and 500 ng of each sequencing ladder was spiked into each experimental sample. The samples were heated to 65 °C for 10 min then transferred to a 96-well plate and frozen for shipping to DNASeq (Dundee, UK) for capillary electrophoresis (CE).

A reverse primer complementary to pgRNA sites 3328–3308 (sequence 3′–5′: AATTTATAAGGGTCAATGTC), was designed to analyse the region encompassing the PS1 signal which was synthesised (IDT) with the appropriate flourophores, (as detailed above) for the aforementioned production of ladders and analysis of beamline-exposed samples.

**CE Analysis**. Normalised reactivity profiles were produced from the raw capillary electrophoresis (CE) data using the QuShape Software package[43] in combination with the BoXFP wrapper module[58]. For each replicate, pre-processing of the CE data was performed, including signal smoothing, decay correction and alignment of peaks in the ddA sequencing trace to corresponding peaks in the XRF trace. Peaks were identified and peaks in different replicates aligned using the molecular weight size marker. Average and standard deviation of peak intensities were calculated across the replicates to produce the intensity profile. Background correction of samples was performed using the average intensity profile across the unexposed background samples computed. The reactivity profile was calculated and normalised, and error propagation was performed to obtain the error on the normalised intensities. Finally, the ddA traces across all samples from the same primer read was used to generate a consensus sequence of U nucleotide locations across the primer read, which was then aligned to the reference genome to determine the position of the primer read in the genome.

Average Pearson correlation coefficients (PCCs) were calculated across the replicates for each treatment (transcript gRNA = 0.8459278, encapsidated gRNA = 0.94918026), and the normalisation factors used in the computation of the normalised reactivity profiles were recorded for each case (transcript gRNA = 2678.25523, NCP gRNA = 1652.53287). The protection ratio across each primer read was calculated as the normalisation factor of the transcript, divided by the normalisation factor for the encapsidated state of the gRNA, calculated as 1.620697097.

**Data acquisition for Cryo-EM**. Lacey carbon 400-mesh copper grids coated with a <3 nm continuous carbon film (Agar Scientific, UK) were glow-discharged in air (10 mA, 30 s) before applying one 3 μL aliquot of pgRNA containing HBV, reassembled and purified as described above for X-Ray Footprinting. Grids were blotted and vitrified in liquid nitrogen-cooled liquid ethane using a LEICA EM GP plunge freezing device (Leica Microsystems). Chamber conditions were set at 4 °C and 95% relative humidity. Grids were stored in liquid nitrogen prior to imaging with an FEI Titan Krios transmission electron microscope (ABSL, University of Leeds) at 300 kV, at a magnification of ×75,000 and a calibrated object sampling of 1.065 Å/pixel. Images were recorded on a FEI Falcon III detector operating in integrating mode. Each movie comprises 59 frames with an exposure rate of 0.9 e− Å$^{-2}$ per frame, with a total exposure time of 1.5 s and an accumulated exposure of 53.1 e− Å$^{-2}$. Data acquisition was performed with EPU Automated Data Acquisition Software for Single Particle Analysis (ThermoFisher) at −0.7 μm to −2.5 μm defocus.

**Image processing**. The established RELION-3.0 pipeline was used for image processing[77,78]. Drift correction was performed using MOTIONCOR2[79], and the contrast transfer function estimated using Gctf[80]. A subset of 1000 particles were picked and classified using reference-free 2D classification and used as templates for the RELION autopicking procedure. Picked particles were sorted using 2D classification, with 127,410 $T = 4$ and 23,257 $T = 3$ particles subsequently extracted. The large majority (~15,000) of $T = 3$ particles were discarded due to heterogeneity. 104,337 $T = 4$ particles were selected and subjected to 3D classification and subsequent auto-refinement with icosahedral symmetry (I1) imposed and without (C1), using EMD-3715 low pass filtered to 60 Å resolution as a reference model. This reconstruction was post-processed to mask and correct for the B-factor of the map. The CTF refinement routine implemented in RELION-3.0[77] was used to refine the reconstruction further, yielding a map with an overall resolution at 3.2 Å, based on the gold-standard (FSC = 0.143) criterion.

To investigate the density of the pgRNA, a focussed 3D classification approach was employed. Each particle contributing to the final icosahedral symmetry-imposed reconstruction was assigned 60 orientations corresponding to its icosahedrally-related views using the relion_symmetry_expand tool. SPIDER[47] was used to generate a spherical mask placed just beneath the Cp shell of the NC, and the symmetry expanded particles were subjected to masked 3D classification, sorting them into 5 classes without alignment, using a regularisation parameter of 25. Particles from these classes were reconstructed using the relion_reconstruct tool, without imposing symmetry, and postprocessed, yielding maps with a resolution of 3.5 Å. UCSF Chimera was used for visualisation and figure generation[81].

**Model building and refinement**. The structure of HBV NC (PDB 3J2V)[53] was first manually docked as a rigid body into the density and followed by real space fitting with the Fit in Map routine in UCSF Chimera[81]. A first step of real space refinement was performed in Phenix[82]. The model was then manually rebuilt in Coot[83] to optimize the fit to the density. After icosahedral symmetrisation to generate the entire capsid, a second step of real space refinement was performed in Phenix. Refinement statistics are listed in Supplementary Table 2.

**Model validation and analysis**. The FSC curve between the final model and map after post-processing in RELION (Model vs Map), is shown in Supplementary Fig. 3b. To perform cross-validation against overfitting, the atoms in the final atomic model were displaced by 0.5 Å in random directions using Phenix. The shifted coordinates were then refined against one of the half-maps (work set) in Phenix using the same procedure as for the refinement of the final model. The other half-map (test set) was not used in refinement for cross-validation. FSC curves of the refined shifted model against the work set (FSCwork) and against the test set (FSCtest), are shown in Supplementary Fig. 3b. The FSCwork and FSCtest curves are not significantly different, consistent with the absence of overfitting in the final models. The quality of the atomic model, including basic protein geometry, Ramachandran plots, clash analysis, was assessed and validated with Coot, MolProbity[84] as implemented in Phenix, and with the Worldwide PDB (wwPDB) OneDep System (https://deposit-pdbe.wwpdb.org/deposition). Graphics were produced by UCSF Chimera[81].

**Reporting summary**. Further information on research design is available in the Nature Research Reporting Summary linked to this article.

## Data availability

The data that supports the findings of this study are available from the corresponding authors upon request. Correspondence and requests for materials should be addressed to N.P or P.G.S. The atomic coordinates for HBV NC were deposited in the Protein Data Bank with code 7ABL. The icosahedrally averaged, symmetry-expanded and focused classified on genome and asymmetric cryo-EM density maps were deposited in the EM Data Bank with codes EMD-11700, EMD-11702 and EMD-11701, respectively.

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

## Acknowledgements

We thank Prof Adam Zlotnick, Indiana University, for the gift of his Cp expression construct and advice on its purification, and Ms Leah Wells who assisted with reassembly experiments during her BSc (Honours) undergraduate research project. We thank the Medical Research Foundation for the award of a career development grant to N.P., and the UK MRC for previous grant funding to study HBV assembly (MRF-044-0002-RG-PATEL & MR/N021517/1). R.T. and P.G.S. thank The Wellcome Trust (Joint Investigator Award Nos. 110145 & 110146 to P.G.S. and R.T., respectively) for funding, and we acknowledge the financial support of The Trust of infrastructure and equipment in the Astbury Centre, University of Leeds (089311/Z/09/Z; 090932/Z/09/Z & 106692), and for their additional support, together with The University of Leeds, of the Astbury Biostructure Facility. R.T. acknowledges additional funding via an EPSRC Established Career Fellowship (EP/R023204/1) and a Royal Society Wolfson Fellowship (RSWF\R1\180009). Portions of this work used the XFP (17-BM) beamline at NSLS-II. Development of XFP was made possible by the National Science Foundation, Division of Biological Infrastructure (grant No. 1228549), while operations support of XFP was provided by the National Institutes of Health (grant No. P30-EB-009998). NSLS-II, a US Department of Energy (DOE) Office of Science User Facility operated for the DOE Office of Science by Brookhaven National Laboratory, was supported under Contract No. DE-SC0012704. We thank DNA Sequencing & Services (MRC I PPU, School of Life Sciences, University of Dundee, Scotland, www.dnaseq.co.uk) for DNA sequencing.

## Author contributions

E.U.W. and S.C. analysed XRF data, C.P.M. helped analyse cryo-EM data and produced structural figures. E.F. and J.B. oversaw experiments at the Brookhaven National Laboratory. N.P. performed experiments and analysed data. N.P. and P.G.S. wrote the paper with help from R.T., N.A.R. and C.P.M. N.P., P.G.S. and R.T. conceived the project.

## Competing interests

The authors declare no competing interests.
