## [Peer Review File · Communications Biology]

Reviewers' comments:

Reviewer #1 (Remarks to the Author):

In this paper, Patel et al follow their 2017 paper in characterizing specific packaging signals (PSs) found in the RNA transcript (pgRNA) encapsidated by the hepatitis B virus (HBV) capsid protein (Core). This time they use an in vitro transcribed pgRNA from one strain and a recombinant Core from another strain. They conduct a series of in vitro encapsidation experiments that lead to formation of icosahedrally ordered nucleocapsid-like particles. In this new in vitro transcribed pgRNA, they mutate the counterparts of three PSs they previously identified. This leads to disruption (3 PSs mutated) or serious impairment (one PS mutated) of packaging and icosahedral nucleocapsid assembly in the particular assay the authors use. They further produce a cryo-EM 3D reconstruction of the reassembled nucleocapsid-like particle. From this and RNA X-ray footprinting data they argue that these PSs, and specifically PS1, are still in contact with Core after assembly. I find that a lot of careful work has gone into this paper, but that the data obtained do not quite support the far-reaching conclusions the authors draw. Specifically:

1) The authors use a T7 in vitro transcript ~3200 nt long as a surrogate for pgRNA, as is usually done for such in vitro assembly experiments. But HBV pgRNA is actually 3500 nt long, capped and polyadenylated. This is a major difference compared to the handful of mutations the authors further introduce.

1-1 The fact that an incomplete pgRNA is used should be mentioned in the introduction and sentences such as (line 96) "we used a full-length pgRNA" or (line 212) "containing the wild-type pgRNA" should be removed and replaced by such as "we used a transcript closer to the size of pgRNA".

1-2 The discussion should also mention that the PSs could be less important than seen here in an actual infection setting, with a bona fide pgRNA and polymerase involved.

1-3 Since they use a 'pgRNA' from one strain and a Core from another, the authors write (line 297) "Evolutionarily conserved PS sites, identified and characterized previously as isolated oligonucleotides, regulate in vitro NCP assembly in the context of a genetically diverse pgRNA strain, with a non-homologous Cp." Fair enough, but from this they cannot conclude (line 300) "The results imply that all HBV strain variants use the same assembly mechanism for production of their nucleocapsid.". Please strongly tone down this statement.

1-4 line 308: "Since this sequence motif is the common feature of the conserved PS sites it suggests that up to sixty(?) similar contacts could occur in NCPs." I do not understand this sentence. There aren't sixty putative PSs in HBV pgRNA. And if the contacts points are at icosahedral axes as the authors suggest (but see below), this is also less than sixty.

2) The cryo-EM work is nice, but quite inconclusive with respect to the author's concern, namely the involvement of PSs:

2-1 Abstract, line 29 "Relaxing the imposed symmetry of this reconstruction reveals a series of pgRNA-core protein contacts associated with the particle symmetry axes" is not correct. Relaxing symmetry did not yield further insights (lines 227-232).

2-2 What the authors actually did (lines 234 seq.) was symmetry expansion of the putative RNA density followed by 3D classification without alignment. In my experience that procedure, while perfectly valid and obviously well done in this case, is prone to artefacts around the symmetry axes. Thus I find it highly suspicious that it led to further density precisely at all symmetry vertices. I would not consider this conclusive evidence.

2-3 Is the "transiting density at the 3-fold vertices" (line 257) also present at the quasi 3-fold ? This would be expected from the quasi-identity of the 3-fold and quasi 3-fold. If not, this is a further clue that the densities seen after symmetry expansion are artefacts.

2-4 Even if this density is real, there is nothing to ascribe it to pgRNA rather than to the C-terminal arginine-rich domain (ARD). Indeed the largest pores of the capsid are the 3-fold and quasi 3-fold axes and likely the pores through which ARDs are extruded during capsid trafficking. Thus the conclusion (line 258) "likely represents part of the pgRNA" is hardly warranted and "and could encompass the PSs described above" is only speculation.

3) I am not convinced either by the RNA X-ray footprinting data, but since this is not my field of expertise I will only point out that:

3-1 By their own admission, the authors had to resort to nonstandard procedures to get the results they were after :line 427 "Ideally, exposed samples would contain no more than 1 cleavage event across the region of interest. However, with RNA genomes as large and variable in structure as the pgRNA of HBV, this was impracticable".

3-2 The results as presented seem at odds with the description that is given, e.g. line 284 "The reactivities of the nucleotides in the stems of these two RNA stem-loops are largely similar as transcript or within the NCP." is not what I see on Fig. 5BC.

4) Misc.

4-1 The discussion is sketchy and seems rushed. Parts of it are actually found in the results, e.g. lines 202-207.

4-2 Similarly in Methods lines 355-368 are actually results. They should be transferred to results or supplementary materials.

4-3 line 310 "consistent with our previous proposal that in HBV pgRNA contacts to the Cp shell help to create a "RNA track" along which the polymerase must move within the NCP". Please cite the paper in which the RNA track model was proposed (to my knowledge): Wang, J.C.-Y., D.G. Nickens, T.B. Lentz, D.D. Loeb, and A. Zlotnick. 2014. Encapsidated hepatitis B virus reverse transcriptase is poised on an ordered RNA lattice. PNAS

4-4 Line 341 "NCPs were dissociated into Cp dimers as previously described". Please indicate that dissociation involves 1.5 M guanidinium chloride.

4-5 Line 397 "Cp aliquots never exceeded 10% of the total volume to limit the final concentration of GuHCl to 0.15 M". Please indicate that (if ?) all samples are adjusted to this final concentration.

4-6 Line 399 "The A260/280 ratio was measured using a Nanodrop™ One (Thermofisher Scientific) and RNA concentration was calculated using absorbance at 260 nm." Does it mean that RNA concentration in NCP solutions was not corrected for light scattering and protein absorbance as in (Porterfield and Zlotnick, Virology 2010) ? In that case the measurements and RNA yield values of Table 1 are certainly overestimated ?

4-7 Undefined abbreviations: ssRNA, GuHCl

Reviewer #2 (Remarks to the Author):

This is an interesting manuscript that studies the role of pgRNA in the assembly of HBV nucleocapsids. The studies are done entirely in vitro with purified HBV Cp and synthetic pgRNA. This is the manuscript's greatest strength and its greatest weakness. The authors find the 3 stem-loop structures in pgRNA facilitate the assembly of HBV capsid in vitro at low concentrations of Cp. They name these sites PS1, PS2, and PS3. A key feature of the three PS sequences is that they can be predicted to form secondary structure that have a RGAG sequence within a loop. The authors find that mutation of all three PS sites or only the PS1 site leads to misassembly and/or incorrect in vitro packaging of pgRNA. Disruption of the PS2 or PS3 site had little to no effect on assembly/packaging. The authors then go on to do a detailed structural analysis and image reconstruction of in vitro assembled particles.

Overall this is an intriguing study. In terms of the biochemistry of HBV capsid assembly and the image reconstruction, it is very impressive. But it is impossible to evaluate whether these findings have any relevance to how HBV replicates in a cell. The roles of these PS sites in the context of viral replication in a cell needs to be performed to evaluate the significance of the finding presented in this manuscript. This is necessary because dogma in the field indicates that the packaging signal, epsilon, is sufficient for packaging of heterologous RNA in a cell as demonstrated in references 29 and 30. None of the packaged mRNA in these studies contain the three PS sequences. Also, it is well established that HBV spliced RNAs, many of which do not contain PS2 and PS3. In addition, Abraham and Loeb (<https://pubmed.ncbi.nlm.nih.gov/17699570>) showed that deletion of the region of the genome containing PS1 had little effect of pgRNA packaging.

To make this study potentially significant, the authors need to place their PS1 mutation in a replication-competent virus genome and measure the level of pgRNA packaging in a cell.

In addition, there are the following concerns that need to be addressed.

The authors claim that the PS site are evolutionarily conserved between different isolates of HBV yet the predicted secondary structures of the 3 PS sequences of two different HBV isolates in figure 2 are not conserved at all. What's up with that?

line 47-48: The evidence and conclusion that Cp associates with cccDNA is controversial. The authors would be best served to trend lightly here.

Line 51-53: HBeAg proteins are not translated from pgRNA. Please correct this statement.

Line 55-57: What is the evidence that T=4 capsids give rise to infectious virions?

Why did the authors choose to do their analysis with the JQ707375.1 genome? Do they have evidence that this genome is replication competent in cells? Why didn't they choose a genome that has been demonstrated to infectious, such as V01460.1?

Reviewer #3 (Remarks to the Author):

Nikesh Patel and colleagues report in this manuscript that specific RNA structure motifs in pgRNA play a critical role in the assembly of HBV nucleocapsids (NCP) in an in vitro assembly assay. Specifically, the authors first identified three mostly conserved pgRNA packaging signals (PS), each of them potentially folds into a stem-loop structure with RGAG sequence in the loop. Mutagenesis analysis on the effects of the three putative PSs in NCP assembly in an in vitro NCP assembly assay revealed that mutation of all three PSs or PS1 alone significantly compromised NCP assembly. However, mutation of PS2 or PS3 did not significantly reduce assembly of NPCs, but the assembled NCPs are RNase-sensitive. Cryo-EM structure reconstruction revealed that the Cp-pgRNA contacts at the 3-fold vertices of NCP around Cp residues E40 and C48. X-Ray RNA footprinting also revealed that the strongest PS of pgRNA, PS1, remains contact with capsid shell within NCPs. Overall, this is a very interesting study and clearly demonstrates the critical role of pgRNA PSs in pgRNA encapsidation/NCP assembly. The conclusions are supported by the results presented.

Specific points

1. Stating from line 53, the sentence "The P, Cp, and HBeAg proteins are translated from the same RNA, a positive-sense, pre-genomic RNA (pgRNA), which also serves as the template for reverse transcription within the nucleocapsid (NC) shell." Is not correct. While the Pol and Cp are translated from pgRNA, pre-core protein is translated from pre-c mRNA and processed into p22 and then p17 to secreted out of hepatocytes as HBeAg.
2. Line 117, does the sentence means that over 90% of pgRNA are packaged into NPCs in this in vitro reaction? How many copies of pgRNA are packaged into a NPC?
3. Line 205, the interpretation and speculations below are not very obvious and should be more clearly elaborated. "It implies that the PSs act cooperatively but differentially during assembly, as seen here³⁷. This mechanism also implies that at least during the assembly step there is a fixed spatial relationship between the pgRNA and Cp shell. This should include a preferred pgRNA conformation within the NCP."

Responses to Reviewers' comments:

Referee expertise:

Referee #1: crystal structure, HBV capsid assembly

Referee #2: hepatitis b virus, capsid assembly, oncology

Referee #3: Molecular pathogenesis, hepatitis B virus, HBV molecular biology

Reviewer #1 (Remarks to the Author):

In this paper, Patel et al follow their 2017 paper in characterizing specific packaging signals (PSs) found in the RNA transcript (pgRNA) encapsidated by the hepatitis B virus (HBV) capsid protein (Core). This time they use an in vitro transcribed pgRNA from one strain and a recombinant Core from another strain. They conduct a series of in vitro encapsidation experiments that lead to formation of icosahedrally ordered nucleocapsid-like particles. In this new in vitro transcribed pgRNA, they mutate the counterparts of three PSs they previously identified. This leads to disruption (3 PSs mutated) or serious impairment (one PS mutated) of packaging and icosahedral nucleocapsid assembly in the particular assay the authors use. They further produce a cryo-EM 3D reconstruction of the reassembled nucleocapsid-like particle. From this and RNA X-ray footprinting data they argue that these PSs, and specifically PS1, are still in contact with Core after assembly.

I find that a lot of careful work has gone into this paper, but that the data obtained do not quite support the far-reaching conclusions the authors draw. Specifically:

1) The authors use a T7 in vitro transcript ~3200 nt long as a surrogate for pgRNA, as is usually done for such in vitro assembly experiments. But HBV pgRNA is actually 3500 nt long, capped and polyadenylated. This is a major difference compared to the handful of mutations the authors further introduce. [This is true, and we compounded the problem by using laboratory shorthand descriptions of the RNA being used. In order to highlight the differences between our minimal molecular system and in vivo viral infection we have added the term "In vitro" to the title, and replaced all appropriate citations of "pgRNA" with "genomic transcript, gRNA etc", highlighting their molecular differences, see Lines 69-70.]

1-1 The fact that an incomplete pgRNA is used should be mentioned in the introduction and sentences such as (line 96) "we used a full-length pgRNA" or (line 212) "containing the wild-type pgRNA" should be removed and replaced by such as "we used a transcript closer to the size of pgRNA". [Agreed, see also above.]

1-2 The discussion should also mention that the PSs could be less important than seen here in an actual infection setting, with a bona fide pgRNA and polymerase involved. [Agreed, see new wording, Lines 339-341.]

1-3 Since they use a 'pgRNA' from one strain and a Core from another, the authors write (line 297) "Evolutionarily conserved PS sites, identified and characterized previously as isolated oligonucleotides, regulate in vitro NCP assembly in the context of a genetically diverse pgRNA strain, with a non-homologous Cp." Fair enough, but from this they cannot conclude (line 300) "The results imply that all HBV strain variants use the same assembly mechanism for production of their

nucleocapsid.". Please strongly tone down this statement. [Agreed, and we have deleted this wording.]

1-4 line 308: "Since this sequence motif is the common feature of the conserved PS sites it suggests that up to sixty(?) similar contacts could occur in NCPs." I do not understand this sentence. There aren't sixty putative PSs in HBV pgRNA. And if the contacts points are at icosahedral axes as the authors suggest (but see below), this is also less than sixty. [Yes we agree, and apologise for this mistake. There are 20 such contacts at 3-folds and a further 20 at the quasi-3-folds, but we have removed the erroneous statement].

2) The cryo-EM work is nice, but quite inconclusive with respect to the author's concern, namely the involvement of PSs: [We accept that at the resolutions we achieved it is impossible to determine whether the density layer we see internal to the globular part of the Cp, and the extensions from it that touch the shell at the symmetry axes, is insufficient to discriminate whether they represent Cp, gRNA or a complex of the two. We have changed our wording accordingly. We do however point out that the density connections at this resolution at 2-fold and 5-fold axes look distinct from each other and from the connections at the 3-folds. Quasi-3-fold and proper 3-fold connections looking similar to each other. The internal layer is at the same radius as the asymmetric feature we saw in NCPs assembled around just PS1. We think it is therefore reasonable to say that this layer may be formed partially of ordered RNA, and have at least shown conclusively that PS1 is in contact with the protein shell via the Cp-recognition motif.]

2-1 Abstract, line 29 "Relaxing the imposed symmetry of this reconstruction reveals a series of pgRNA-core protein contacts associated with the particle symmetry axes" is not correct. Relaxing symmetry did not yield further insights (lines 227-232). [Again apologies for this misstatement, which has been corrected.]

2-2 What the authors actually did (lines 234 seq.) was symmetry expansion of the putative RNA density followed by 3D classification without alignment. In my experience that procedure, while perfectly valid and obviously well done in this case, is prone to artefacts around the symmetry axes. Thus I find it highly suspicious that it led to further density precisely at all symmetry vertices. I would not consider this conclusive evidence. [We apologise for not making this clearer in the original submission. In order to clarify these issues we have replaced Figure 4, that now allows the reader to compare the transiting densities at each of the vertices. These density connections at both 3-folds and quasi-3-folds seem quite similar.]

2-3 Is the "transiting density at the 3-fold vertices" (line 257) also present at the quasi 3-fold ? This would be expected from the quasi-identity of the 3-fold and quasi 3-fold. If not, this is a further clue that the densities seen after symmetry expansion are artefacts. [See above.]

2-4 Even if this density is real, there is nothing to ascribe it to pgRNA rather than to the C-terminal arginine-rich domain (ARD). Indeed the largest pores of the capsid are the 3-fold and quasi 3-fold axes and likely the pores through which ARDs are extruded during capsid trafficking. Thus the conclusion (line 258) "likely represents part of the pgRNA" is hardly warranted and "and could encompass the PSs described above" is only speculation. [Yes, we have changed the language to make clear that we are assuming that to have the effects seen on in vitro assembly the transiting densities might encompass PS RNA. We then test that proposal, i.e. that there is Cp-PS RNA contact directly via XRF, making it clear we hope that this does not prove the contact corresponds to the transiting density.]

3) I am not convinced either by the RNA X-ray footprinting data, but since this is not my field of expertise I will only point out that:

3-1 By their own admission, the authors had to resort to nonstandard procedures to get the results

they were after :line 427 "Ideally, exposed samples would contain no more than 1 cleavage event across the region of interest. However, with RNA genomes as large and variable in structure as the pgRNA of HBV, this was impracticable". [The initial procedure we were following was formulated using rRNA in solution. Viral genomes turn out to be different, and in this case all samples were frozen and the assumption we made is a legitimate one. We know this is the case for bacteriophage MS2. We made the same assumption there and got the same secondary structure for its gRNA as seen at atomic resolution in asymmetric cryo-EM. We have altered the wording to make it clear that this is now a validated assumption, and included a reference to our manuscript explaining all this on Biorxiv (ref #59).]

3-2 The results as presented seem at odds with the description that is given, e.g. line 284 "The reactivities of the nucleotides in the stems of these two RNA stem-loops are largely similar as transcript or within the NCP." is not what I see on Fig. 5BC. [We agree that the present colour-coding, which is common in the foot-printing field, is not as clear as it could be. We have added two forms of the per nucleotide XRF reactivity differences in this region to Fig 5C & D in the hope that this is clearer.]

4) Misc.

4-1 The discussion is sketchy and seems rushed. Parts of it are actually found in the results, e.g. lines 202-207. [We have tried to provide fuller descriptions and eliminate repetitions throughout.]

4-2 Similarly in Methods lines 355-368 are actually results. They should be transferred to results or supplementary materials. [See response to 4.1]

4-3 line 310 "consistent with our previous proposal that in HBV pgRNA contacts to the Cp shell help to create a "RNA track" along which the polymerase must move within the NCP". Please cite the paper in which the RNA track model was proposed (to my knowledge): Wang, J.C.-Y., D.G. Nickens, T.B. Lentz, D.D. Loeb, and A. Zlotnick. 2014. Encapsidated hepatitis B virus reverse transcriptase is poised on an ordered RNA lattice. PNAS [Thank you for spotting our failure to cite this original proposal from the Zlotnick group. Their paper is now cited as reference 50.]

4-4 Line 341 "NCPs were dissociated into Cp dimers as previously described". Please indicate that dissociation involves 1.5 M guanidinium chloride. [done, Line 124].

4-5 Line 397 "Cp aliquots never exceeded 10% of the total volume to limit the final concentration of GuHCl to 0.15 M". Please indicate that (if ?) all samples are adjusted to this final concentration. [All sample aliquots end up at the same 0.15 M GuHCl concentration, and this is now specified in the text, Line 423.]

4-6 Line 399 "The A260/280 ratio was measured using a Nanodrop™ One (Thermofisher Scientific) and RNA concentration was calculated using absorbance at 260 nm." Does it mean that RNA concentration in NCP solutions was not corrected for light scattering and protein absorbance as in (Porterfield and Zlotnick, Virology 2010)? In that case the measurements and RNA yield values of Table 1 are certainly overestimated? [Agreed, we have now applied the appropriate corrections.]

4-7 Undefined abbreviations: ssRNA, GuHCl [defined, see lines 50 & 124.]

Reviewer #2 (Remarks to the Author):

This is an interesting manuscript that studies the role of pgRNA in the assembly of HBV nucleocapsids. The studies are done entirely in vitro with purified HBV Cp and synthetic pgRNA. This is the manuscript's greatest strength and its greatest weakness. The authors find the 3 stem-loop structures in pgRNA facilitate the assembly of HBV capsid in vitro at low concentrations of Cp. They name these sites PS1, PS2, and PS3. A key feature of the three PS sequences is that they can be predicted to form secondary structure that have a RGAG sequence within a loop. The authors find that mutation of all three PS sites or only the PS1 site leads to misassembly and/or incorrect in vitro packaging of pgRNA. Disruption of the PS2 or PS3 site had little to no effect on assembly/packaging. The authors then go on to do a detailed structural analysis and image reconstruction of in vitro assembled particles.

Overall this is an intriguing study. In terms of the biochemistry of HBV capsid assembly and the image reconstruction, it is very impressive. But it is impossible to evaluate whether these findings have any relevance to how HBV replicates in a cell. The roles of these PS sites in the context of viral replication in a cell needs to be performed to evaluate the significance of the finding presented in this manuscript. This is necessary because dogma in the field indicates that the packaging signal, epsilon, is sufficient for packaging of heterologous RNA in a cell as demonstrated in references 29 and 30. None of the packaged mRNAs in these studies contain the three PS sequences. Also, it is well established that HBV spliced RNAs, many of which do not contain PS2 and PS3. In addition, Abraham and Loeb (<https://pubmed.ncbi.nlm.nih.gov/17699570>) showed that deletion of the region of the genome containing PS1 had little effect on pgRNA packaging.

To make this study potentially significant, the authors need to place their PS1 mutation in a replication-competent virus genome and measure the level of pgRNA packaging in a cell. [With respect, we feel that such experiments can be misleading for the reasons set out above. The fragment carrying epsilon in the references you cite also contains previously unsuspected PSs, as we now describe in the Sup. Info. Heterologous RNAs that were added to it do not contain multiple PSs, and so their combination, not simply the presence of the epsilon sequence we would predict should lead to assembly, as it does. We are cognizant of your more general points about the differences between in vitro and in vivo studies however. We have tried therefore to emphasise that our work is with a minimal molecular system in vitro, but if the same sequences of Cp and pgRNA are present in vivo we would expect that they would also behave similarly, and that therefore PS-mediated assembly could contribute to in vivo assembly.]

In addition, there are the following concerns that need to be addressed.

The authors claim that the PS sites are evolutionarily conserved between different isolates of HBV yet the predicted secondary structures of the 3 PS sequences of two different HBV isolates in figure 2 are not conserved at all. What's up with that? [Sorry, the language was not very clear. The conservation of these three PS sites was determined by the matching of millions of anti-Cp aptamer reads against database genome sequences, as described in Patel et al, ref #12.] It was not meant to imply that PS sites are all identical, indeed variation around a consensus (Line 210) is needed to define a hierarchy of Cp affinity, and hence a preferred assembly path with respect to the pgRNA. We have tried to make this clearer by discussing the variation of PS sites here; Lines 93-96.]

line 47-48: The evidence and conclusion that Cp associates with cccDNA is controversial. The authors would be best served to trend lightly here. [Thank you for this note of caution. We have changed the wording accordingly, Line 48.]

Line 51-53: HBeAg proteins are not translated from pgRNA. Please correct this statement. [The language has been changed to indicate that only Pol and Cp are transcribed from the pgRNA – Line 54].

Line 55-57: What is the evidence that T=4 capsids give rise to infectious virions? [Good question, we've deleted this statement.]

Why did the authors choose to do their analysis with the JQ707375.1 genome? Do they have evidence that this genome is replication competent in cells? Why didn't they choose a genome that has been demonstrated to be infectious, such as V01460.1? [JQ707375.1 was chosen because it was commercially available and fuller details on this are included at Lines 98-108.]

Reviewer #3 (Remarks to the Author):

Nikesh Patel and colleagues report in this manuscript that specific RNA structure motifs in pgRNA play a critical role in the assembly of HBV nucleocapsids (NCP) in an in vitro assembly assay. Specifically, the authors first identified three mostly conserved pgRNA packaging signals (PS), each of them potentially folds into a stem-loop structure with RGAG sequence in the loop. Mutagenesis analysis on the effects of the three putative PSs in NCP assembly in an in vitro NCP assembly assay revealed that mutation of all three PSs or PS1 alone significantly compromised NCP assembly. However, mutation of PS2 or PS3 did not significantly reduce assembly of NCPs, but the assembled NCPs are RNase-sensitive. Cryo-EM structure reconstruction revealed that the Cp-pgRNA contacts at the 3-fold vertices of NCP around Cp residues E40 and C48. X-Ray RNA footprinting also revealed that the strongest PS of pgRNA, PS1, remains contact with capsid shell within NCPs. Overall, this is a very interesting study and clearly demonstrates the critical role of pgRNA PSs in pgRNA encapsidation/NCP assembly. The conclusions are supported by the results presented.

Specific points

1. Stating from line 53, the sentence “The P, Cp, and HBeAg proteins are translated from the same RNA, a positive-sense, pre-genomic RNA (pgRNA), which also serves as the template for reverse transcription within the nucleocapsid (NC) shell.” Is not correct. While the Pol and Cp are translated from pgRNA, pre-core protein is translated from pre-c mRNA and processed into p22 and then p17 to secreted out of hepatocytes as HBeAg. [Thank you for spotting this error. The language has been changed to indicate that only Pol and Cp are transcribed from the pgRNA – Line 54].
2. Line 117, does the sentence means that over 90% of pgRNA are packaged into NCPs in this in vitro reaction? How many copies of pgRNA are packaged into a NCP? [Our assumption is that assembly of the wild-type transcript is highly efficient under these conditions based on a) the $A_{260/280}$ values of the NCPs produced; b) that they are the same size as bona fide NCPs judged by the gel-filtration/SEC-Malls; and c) all particles exclude negative stain from their interiors. The implications are that the vast majority of the gRNA transcript molecules assemble into NCPs that have a single RNA in each. Please see new text at Lines 129-134.]
3. Line 205, the interpretation and speculations below are not very obvious and should be more clearly elaborated. “It implies that the PSs act cooperatively but differentially during assembly, as seen here³⁷. This mechanism also implies that at least during the assembly step there is a fixed spatial relationship between the pgRNA and Cp shell. This should include a preferred pgRNA conformation within the NCP.” [We have tried to make this section clearer and in particular we have cited the primary papers on which these assumptions are based. Please see new text at Lines 209-219.]

REVIEWERS' COMMENTS:

Reviewer #1 (Remarks to the Author):

In this revision, Patel et al. have carefully addressed my concerns, namely toning down their bolder statements and revising the presentation of their cryo-EM analyses (e.g. their new Fig. 4). I may add that I feel the toning down is enough to answer the concerns of Reviewer #2. The paper is now clearly identified as in vitro work providing further evidence for the involvement of PSs in HBV nucleocapsid assembly. Testing its conclusions in HBV replication systems is quite beyond its scope.

As to the cryo-EM work, it is reassuring that they find very similar connections at 3-fold and quasi-3-fold axes after focused classification and asymmetric reconstruction. I would still request the authors to change

line 29 'Symmetry relaxation of this reconstruction reveals that such contacts are made at every symmetry axis.'

to something like

'Asymmetric reconstructions reveal that such contacts are made at every symmetry axis.' which better alerts the reader to the fact that special image analyses, not just symmetry relaxation, were performed to visualise these contacts.

Reviewer #3 (Remarks to the Author):

The authors fully addressed my concerns on the previous version of the manuscript with satisfaction.